# *Bacillus velezensis* Strain GUMT319 Reshapes Soil Microbiome Biodiversity and Increases Grape Yields

**DOI:** 10.3390/biology11101486

**Published:** 2022-10-11

**Authors:** Xiangru Chen, Fang Yang, Chunwei Bai, Qianrui Shi, Shan Hu, Xianying Tang, Lijuan Peng, Haixia Ding

**Affiliations:** 1Department of Plant Pathology, College of Agriculture, Guizhou University, Guiyang 550025, China; 2Sino Green Agri-Biotech Co., Ltd., Beijing 102101, China; 3Tonghe Zhiyuan (Beijing) Environmental Protection Technology Co., Ltd., Beijing 100036, China; 4Liangshan Yi Autonomous Prefecture Bureau of Agriculture and Rural Affairs, Xichang 615000, China; 5College of Tobacco Science, Guizhou University, Guiyang 550025, China

**Keywords:** *Bacillus velezensis*, PGPR, *Vitis vinifera*, rhizobacteria

## Abstract

**Simple Summary:**

This article confirms that *Bacillus velezensis* strain GUMT319 is a plant growth-promoting rihizobacteria that uses GUMT319 to increase the yield of grape. To reveal the mechanism of GUMT319 in increasing the yield of grape, we analyzed the soil property and soil microbial community composition of GUMT319-treated and untreated soil. Results showed that the physical and chemical properties and the microbial composition of soil were all altered by *Bacillus velezensis* strain GUMT319.

**Abstract:**

*Bacillus velezensis* strain GUMT319 is a rhizobacteria biocontrol agent that can control tobacco black shank disease. We took GUMT319 as a biological fertilizer on *Vitis vinifera* L. The test group was treated with GUMT319 for one year and the control group had a water treatment. Yields of GUMT319-treated grape groups were significantly increased compared to the controls. The average length and width of single grape fruit, weight of 100 grape fruits, the sugar/acid ratio, and the content of vitamin C were all increased in the GUMT319-treated grape group. The pH of the soil was higher and the contents of alkaline hydrolyzable nitrogen and available potassium were significantly lower in the GUMT319-treated groups than the controls. The soil microbial community composition was evaluated by 16S rDNA high-throughput sequencing, and the Shannon index and Simpson index all showed that soil microbes were more abundant in the GUMT319-treated group. These results indicate that GUMT319 is not only a biocontrol agent, but also a plant growth-promoting rihizobacteria. It can increase the yield of grape by altering the physical and chemical properties and the microbial community composition of the soil.

## 1. Introduction

Rhizosphere bacteria include many beneficial microorganisms. Rhizosphere bacteria that increase the yield and quality of plants have been termed plant growth-promoting rhizobacteria (PGPR) [1]. So far, many PGPR have been identified. For example, *Pseudomonas fluorescens* strain WCS417r can prevent the invasion of *Fusarium. oxysporum* by inducing the systemic resistance of carnation plants [2]; *P. fluorescens* strain G8-4 and *P. putida* strain 34-13 can inhibit the infection of *Colletotrichum orbiculare* by inducing the systemic resistance of cucumber [3]; *P. fluorescens* strains MKB 100 and MKB 249, *P. frederiksbergensis* strain 202, and *Pseudomonas* sp. strain MKB 158 can control *Fusarium* spp. in wheat and barley by secreting antibiotic substances [4]; *P. putida* strain 06,909 can significantly reduce cadmium phytotoxicity and increase the metal accumulation in sunflower plant roots [5]; nitrogen-fixing rhizobacteria can increase the nitrogen content in legume host plants, for example *Rhizobium etli* can increase the nitrogen content in beans [6]; after treatment by *B. circulans,* the total tuber yields and average tuber weight of per treated potato were significantly higher than in the control group [7].

The mechanisms of how PGPR improve the yield and quality of plants are well studied and have been summarized in a recent review paper. PGPR functions generally consist of biocontrol, enhancement of plant tolerance, and plant growth promotion [8]. These aspects can be further divided into components. For example, the PGPR mechanisms used to promote plant growth can vary. *P. stutzeri* strain A1501 can increase the growth and nitrogen accumulation of maize [9]. At least 50% of all bacteria can solubilize phosphorus. *Bacillus*, *Rhizobium,* and *Pseudomonas* are the most significant phosphorus-solubilizing bacteria [10]. *B**acillus mucilaginosus*, *B. edaphicus*, and *B. circulans* have efficient potassium-solubilizing capability [11]. Some PGPR species promote the iron uptake ability of plant roots and reduce the abundance of pathogenic microbes through the production of siderophores [12,13]. *P. aeruginosa* strain PF23EPS+ can increase the production of salicylic acid in sunflower and promote plant growth [14]. The colonization of PGPR in the rhizosphere and roots can exclude plant disease pathogens and help plants absorb soil nutrients [15].

Bacilli rhizobacteria is a potential repository for PGPR [16,17]. Many *Bacillus* spp. are reported to be PGPR. Colonization of *B. velezensis* strain SQR9 on cucumber roots can increase root secretion of tryptophan to increase colonization of SQR9 and reduce raffinose secretion of cucumber roots. This can inhibit the colonization of *F. oxysporum* f. sp. *cucumerinum* [18,19]. *B. velezensis* strain FZB42 has been considered to be a Gram-positive model strain for studies on plant growth promotion and biocontrol [20,21]. The subclass IId bacteriocin thuricin 17 produced by *B. thuringiensis* strain NEB17 stimulates plant growth against abiotic stresses [22]. Biological and bioinformatical tests demonstrated that *B. velezensis* strain SC60 can inhibit the growth of many plant pathogens and promote the growth of *Sesbania cannabina* [23].

The overuse of fertilizers and pesticides has caused many problems for agricultural production and the environment. Application of PGPR can help maintain sustainable agricultural development [8,24,25,26,27]. *B. velezensis* strain GUMT319 is a PGPR rhizobacteria isolated from Guizhou province [28]. In this study, we treated vineyard soils with GUMT319 to determine the effects of GUMT319 on grape yield and found that GUMT319 can function like a PGPR.

## 2. Materials and Methods

### 2.1. Experimental Design and Plant Management

The *Bacillus velezensis* strain GUMT319 rhizobacteria was originally isolated from the rhizosphere of field-grown tobacco plants in Guizhou with a high incidence of tobacco black shank. Previous studies confirmed that GUMT319 is a useful biocontrol agent [28]. We treated the *Vitis vinifera* L. variety ‘Sweet Sapphire’ with GUMT319 by root irrigation to determine its impact on grape plant growth. We set up three experimental replicates with an area of 0.667 hm^2^ each for both the GUMT319-treated group and the GUMT319 untreated group. During growth, the grape plants were irrigated with 3 kg of a GUMT319-containing powder (amount of GUMT319 was 3 × 10^1^^1^ CFU/g) (CFU stands for colony forming units) per 0.0667 hm^2^ to the rhizosphere of grape plants. The treatment frequency of GUMT319 on grape plants was every four months from 10 October 2019 to 12 October 2020. The land areas of the GUMT319-treated and untreated group are both 0.667 hm^2^, and two sets of biological replicates were set up in the same garden. All the other agricultural practices were the same between the GUMT319-treated and untreated grape plants.

### 2.2. Physical and Chemical Properties Evaluation of the Grape

One hundred grape fruits of each treatment were randomly picked and evaluated. After weighing, the grape fruits were measured for length (polar) and width (equatorial) of each fruit (mm) using a Vernier caliper [29]. The firmness of grape fruits were tested as the method described previously [30]. Ten fruits of each treatment were randomly picked and used to analyze flesh firmness with a penetrometer (Model FT327; Effegi). Each grape fruit was analyzed three times from three sites (without peel) along the equator, with approximately 120° apart.

Vitamin C content in grapes was determined by molybdenum blue colorimetry [31]. The sugar and acid content of the grapes were evaluated using headspace solid phase micro-extraction/gas chromatography–mass spectrometry (HS-SPME/GC-MS) analysis method proposed by Song et al., 2019 [32].

### 2.3. Soil Sampling

In August 2021, 10 months after the last irrigation of GUMT319, soil samples (1 kg each) were randomly collected from locations in the grape orchard at a depth of 0–20 cm near the rhizosphere of GUMT319-treated or untreated grape plants. We collected the soil samples 10 months after the last treatment of GUMT319, because we wanted to eradicate the effects of GUMT319 itself on the 16S rDNA sequencing result. Three samples from the GUMT319-treated area and three samples from the untreated area were collected. The soil samples were taken to a laboratory for measurement of soil chemical properties and for 16S rDNA high-throughput sequencing.

### 2.4. Determination of Soil Chemical Properties

The soil samples were sent to the Agricultural Resources and Environment Center of Guizhou Academy of Agricultural Sciences (ARECGAAS) where the chemical properties were measured. The soil pH was determined using a method previously reported [33]. Suspensions (1:2.5 soil: CaCl_2_) were prepared and the pH was measured using a laboratory 870 pH meter (Schott Instruments, Mainz, Germany) [34]. The organic matter (C, N, and S) and elemental composition (wt %) in soil samples were determined following the method established by Anderson [35].

### 2.5. 16S rDNA Sequencing and Bioinformatics

The 16S rDNA sequencing followed previously described protocols [36,37]. The V3-V4 region of 16S rDNA was cloned and prepared for high-throughput sequencing on an Illumina HiSeq 2500 platform in Novogene Co., Ltd. (Novogene, Beijing, China). The DADA2 method is mainly used for noise reduction. The DADA2 method is more sensitive and specific than the traditional operational taxonomic units (OTU) method and can detect the real biological variation missed by the OTU method while producing fewer false sequences [38]. QIIME2 and QIIME were used separately to obtain the alpha-diversity and beta-diversity of all groups [39,40]. Simpson and Shannon indices were used to identify the richness and diversity of the microbiome of each group [41]. In the Beta diversity research, four indicators, including the weighted unifrac distance, unweighted unifrac distance, jaccard distance, and Bray–Curtis distance, were used to measure the dissimilarity coefficient of the two samples. Then, principal coordinates analysis (PCoA) and principal component analysis (PCA) were conducted to evaluate the differences between the sample groups [42,43]. Differences and changes related to functional genes in the microbial community of different groups of samples in the metabolic pathway were observed through the composition and difference analysis of the Kyoto Encyclopedia of Genes and Genomes (KEGG) metabolic pathway using PICRUSt2 software.

### 2.6. Statistical Analysis

Data in this study were analyzed by SPSS and Origin [44]. One-way analysis of variance was used to compare the data, followed by Duncan’s test at 5% level of significance to determine the significance of differences between treatment means. All analyzed data are presented as arithmetic means ± standard deviation [45].

## 3. Results

### 3.1. Effect of B. velezensis Strain GUMT319 on Grape Growth

Results of the field experiment are shown in Figure 1. After one year of constant treatment, the morphology of the GUMT319-inoculated grape plants was significantly different from the control plants. The ear length of the GUMT319-treated grape plants is longer than the controls, and the roots of GUMT319-treated plants had more branches and root hairs than the control plants.

### 3.2. Fruit Morphological and Architectural Parameters

To evaluate the effect of GUMT319 on grape fruits, we measured physiological indicators of both the GUMT319-treated and control grapes. The lengths and widths of the grapes in the treatment group were significantly larger than those in the control group (Figure 2A,B). The firmness and the weight of a single fruit were also measured. Fruits of GUMT319-treated plants had greater firmness and higher 100-fruit weight than fruit from the control plants (Figure 2C,D). We examined the sugar/acid ratio and the vitamin C content of the fruits and found that the sugar/acid ratio and vitamin C content of grapes in the GUMT319-treated group were significantly higher than those in the control group (Figure 2E,F).

### 3.3. Changes in Soil Chemical Properties

Because the growth of plants is inseparable from the soil environment in which they are located, we measured the physicochemical properties of control soil and soil treated with GUMT319. The soil pH of the GUMT319 treatment was significantly higher than that of the control group (Figure 3A). The levels of alkaline hydrolyzable nitrogen and available potassium in the treatment group were significantly lower than those in the control group (Figure 3B,C). There were no significant changes in soil organic matter, total potassium, and total phosphorus content (Figure 3D–F).

### 3.4. Effect of GUMT319 on the Soil Microbial Community Composition

The change of rhizosphere microbial composition also affects the physical and chemical properties of soil and the growth of plants. Therefore, we conducted 16S rDNA high-throughput sequencing on the soil sample of GUMT319-treated and untreated plants to observe the impact of GUMT319 on the rhizosphere microbial composition. The 16S rDNA sequencing results showed differences in the composition of grape rhizosphere microbes between the GUMT319 treated and non-treated plants; there are 37 upregulated microbes and 6 downregulated microbes in GUMT319-treated soil when compared to the untreated soil (Figure 4). Both the Shannon index (Figure 5A) and the Simpson index (Figure 5B) results confirmed that the species richness of rhizosphere microorganisms in the GUMT319-treated soil was higher than that in the untreated soil. We also performed a functionally annotated relative abundance cluster analysis of rhizosphere microbes between the GUMT319-treated and untreated soil. Results showed that genes in the KEGG pathways, such as KO02529, KO05349, KO2552, and KO2454, were expressed differently in GUMT319-treated and untreated soil microorganisms (Figure 6).

## 4. Discussion

Providing adequate food for the continuously increasing human population is a worldwide problem [46]. Increased use of fertilizers and pesticides provides some benefits, but their use is accompanied by many environmental problems. Therefore, more environmentally friendly and sustainable ways for increasing food production are needed. PGPR microorganisms were proved to have the ability to enhance yield in many crop production systems. For example, *P. fluorescens* N04, *P. koreensis* N19, *Paenibacillus alvei* T19, and *Lysinibacillus sphaericus* T22 can alter the synthesis of secondary metabolites and aromatic amino acids of tomato plants to increase their defense response to various stresses (either biotic or abiotic) and increase fruit yields [47]. The endophytic strain of *Beauveria bassiana* can promote the growth of grape (*Vitis vinifera*), but the mechanism by which *B. bassiana* functions to promote grape growth is unclear [48].

We found that the *B. velezensis* strain GUMT319, a newly identified biocontrol agent for controlling tobacco black shank disease, also has a yield-increasing effect on grapes; the fruits on GUMT319-treated grape plants were larger and heavier than fruits on control grape plants. In addition, the sugar/acid ratio and vitamin C content of the GUMT319-treated grapes were higher than the controls. The ideal substrate for plant growth usually has a high water-holding capacity, good rehydration after drying, good aeration, stable structure, optimal pH, high cation exchange capacity, absence of toxic compounds, and low microbial activity and is free of pests and weeds [49]. We tested the chemical properties of the GUMT319-treated soils to determine if the yield-improving action of GUMT319 on grape plants functions in altering the physical and chemical properties of the soils. We found that the soil pH value increased from 4.95 to 5.80 after GUMT319 treatment, and the levels of alkaline hydrolyzable nitrogen and available potassium were reduced in the GUMT319-treated field. The microbial community can increase the storage of C in near-neutral pH soils [50]. The increase of C in the soil may be related to the increased yield of GUMT319-treated grape plants; however, this is only one of our inferences, and further experiments need to be designed to prove whether this view is correct or not. The decrease in alkaline hydrolyzable nitrogen and available potassium in the GUMT319 treatment suggests that the grapes treated with GUMT319 absorbed more alkaline hydrolyzable nitrogen and available potassium from the soil. The alkaline hydrolyzable nitrogen and available potassium were beneficial to the synthesis of organic matter in all organisms, including the grape plants; this might also be the reason why the grape plants treated with GUMT319 could have higher yield. 

We further compared the composition of the rhizosphere microbial community of the GUMT319-treated soil and untreated control soil by 16S rDNA high-throughput sequencing. Result showed that the rhizosphere microbial composition of soil treated with GUMT319 was significantly more abundant than the untreated soil. Microbes in GUMT319-treated and untreated soil showed functional differences in KEGG pathway like KO02529, KO05349, KO2552, and KO2454. These differently expressed microbial community and KEGG pathways could also associated with the altered yield of grape plants; for example, KO2552 is a pathway that increased in GUMT319-treated soil. It is a pathway that directs the decomposition of N(alpha)-Benzyloxycarbonyl-L-leucine into Benzyl alcohol, CO_2_, and L-Leucine (https://www.kegg.jp/entry/R02552, accessed on 1 October 2022). As we know leucine is an important amino acid for plants and is required for plant growth, it is very logical to think about whether the increase of leucine sources in soil would affect the yield of grape, but this hypothesis need further experiments to verify it.

## 5. Conclusions

In summary, this study demonstrated that the *Bacillus velezensis* strain GUMT319 is a PGPR. It can significantly increase grape yields, and it can alter the soil microbial community composition. The change of the soil microbial community may connect with the yield increase and the alternation of the physical and chemical properties of the soil, but these hypotheses need to be further studied.

## Figures and Tables

**Figure 1 biology-11-01486-f001:**
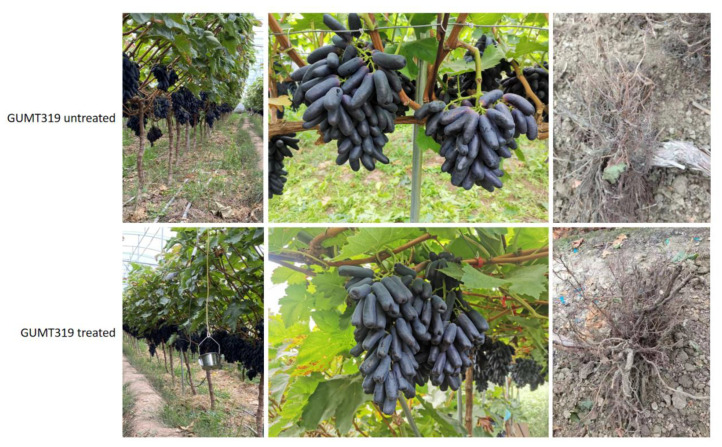
Growth and yield parameters of grape plants treated with and without *Bacillus velezensis* strain GUMT319.

**Figure 2 biology-11-01486-f002:**
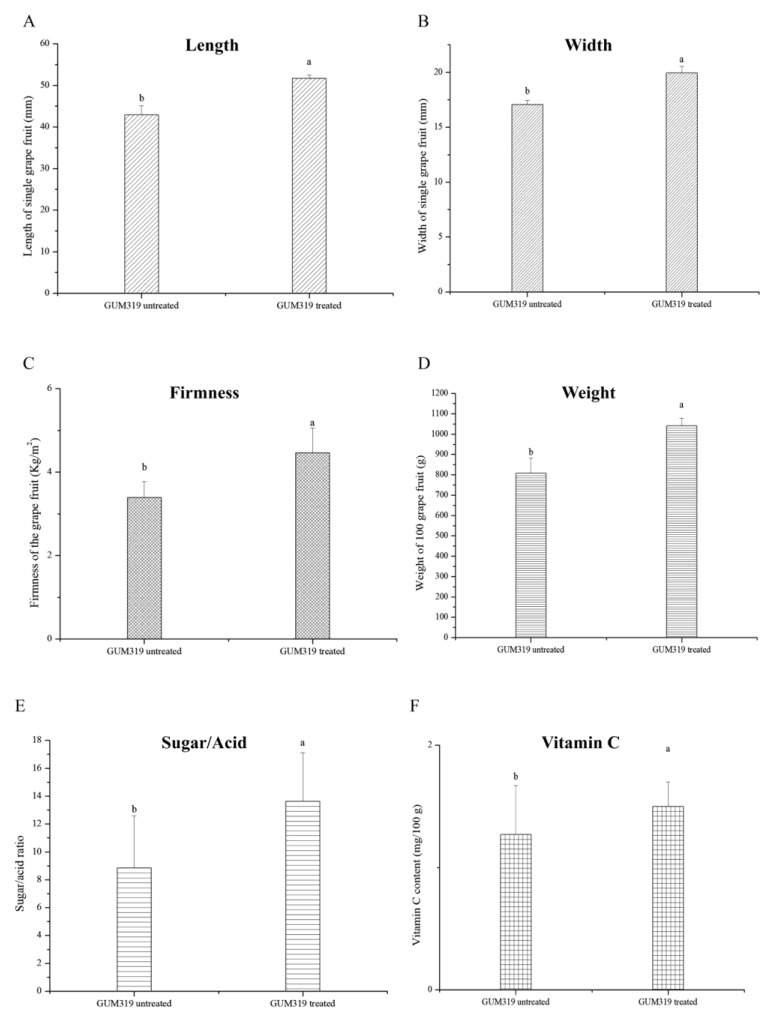
Grape fruit parameters. The error bars represent standard deviation (SD). (**A**) Length of single fruit: GUMT319 untreated 42.94 ± 2.19 b, and GUMT319 treated 51.68 ± 0.83 a (mm). (**B**) Width of single fruit: GUMT319 untreated 17.06 ± 0.39 b, and GUMT319 treated 19.94 ± 0.59 a (mm). (**C**) Firmness of grape fruit: GUMT319 untreated 3.39 ± 0.38 b, and GUMT319 treated 4.46 ± 0.60 a (Kg/m^2^). (**D**) Weight of 100 grape fruits: GUMT319 untreated 808.03 ± 74.73 b, and GUMT319 treated 1040.87 ± 37.42 a. (**E**) Sugar/acid ratio: GUMT319 untreated 8.85 ± 4.23 b, and GUMT319 treated 13.64 ± 3.48 a. (**F**) The content of vitamin C: GUMT319 untreated 1.27 ± 0.40 b, and GUMT319 treated 1.50 ± 0.20 a.

**Figure 3 biology-11-01486-f003:**
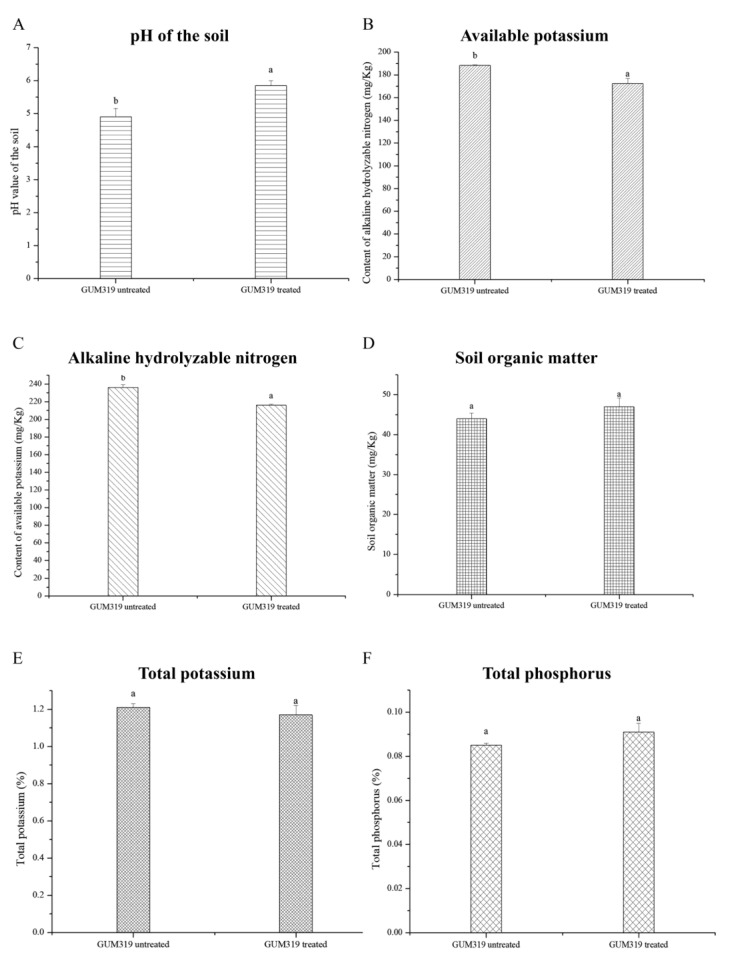
Properties of the GUMT319-treated and untreated soil. The error bars represent standard deviation (SD). (**A**) The pH value of the soil: GUMT319 untreated 4.90 ± 0.26 b, and GUMT319 treated 5.85 ± 0.15 a. (**B**) The content of available potassium: GUMT319 untreated 188.33 ± 0.47 b, and GUMT319 treated 172.33 ± 4.71 b (mg/Kg) (**C**) The content of alkaline hydrolyzable nitrogen: GUMT319 untreated 236.16 ± 3.00 b, and GUMT319 treated 215.94 ± 1.32 a (mg/Kg). (**D**) The content of soil organic matter: GUMT319 untreated 47 ± 2.22 a, and GUMT319 treated 47 ± 1.43 a (mg/Kg). (**E**) The ratio of total potassium: GUMT319 untreated 1.21 ± 0.02 a, and GUMT319 treated 1.17 ± 0.05 a (%). (**F**) The ratio of total phosphorus: GUMT319 untreated 0.085 ± 0.001 a, and GUMT319 treated 0.091 ± 0.004 a (%).

**Figure 4 biology-11-01486-f004:**
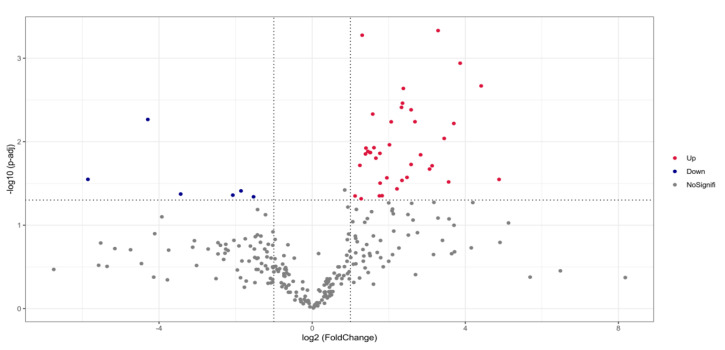
Volcano map of the species diversity of the microbes in GUMT319-treated and untreated grape plant rhizosphere. Result showed that there are 37 upregulated microbes and 6 downregulated microbes that appeared in GUMT319-treated soil compared to the untreated soil. Species with differences in richness were selected by log2 fold change > 1 and q value < 0.005. NoSignifi represents microbes with no significant difference in species richness.

**Figure 5 biology-11-01486-f005:**
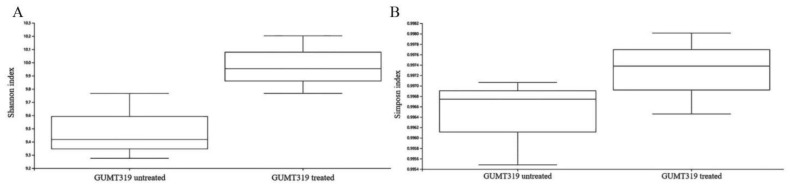
Species diversity analysis of the microbes in GUMT319-treated and untreated grape plant rhizosphere. Both (**A**) Shannon index and (**B**) Simpson index results showed that the species richness of rhizosphere microorganisms in the GUMT319 treatment groups was higher than that in the untreated group.

**Figure 6 biology-11-01486-f006:**
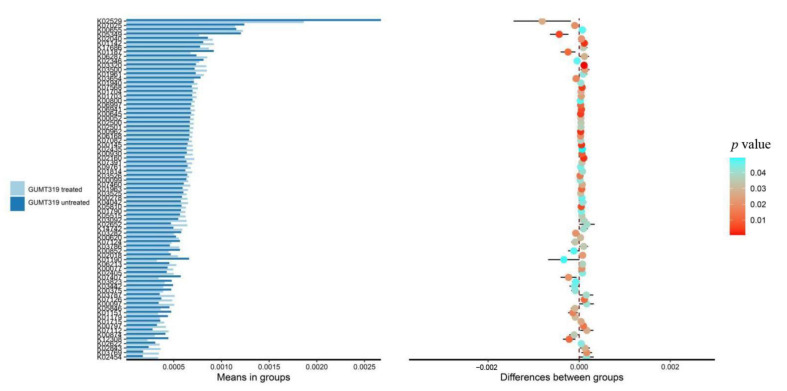
Differences and changes related to functional genes in the microbial community of GUMT319-treated and untreated samples in the metabolic pathway.

## Data Availability

All data can be available from the corresponding author.

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
