# Peer review of "Bacillus velezensis Strain GUMT319 Reshapes Soil Microbiome Biodiversity and Increases Grape Yields"

_biology, 2022, doi:10.3390/biology11101486_

Round 1
Reviewer 1 Report
The manuscript entitled “Bacillus velezensis strain GUMT319 reshapes microbiome biodiversity and increases grape yields” lacks a solid experimental design that would support the conclusions it proposes. Likewise, the reported methodology is incomplete, and, in some cases, it does not even seem adequate to evaluate the responses reported. Thus, the conclusions of the manuscript are not supported, and the manuscript should not be considered for publication in Biology.
Specific Comments
The authors study the potential of B.velezensis GUMT319 as PGPR, having as main background the work of Ding et al., 2021 (lines 78 and 85). This reference, which appears to be from the same research group, is not included in the bibliography.
Lines 87 to 88 mention the applied dose of GUMT319 (≈ 4 x 10 15 CFU/Ha) while the commercial doses of PGPR are in the order of 2 to 4 x 10 12 CFU/Ha. The foregoing is that the activity and/or permanence of the bacteria in the rhizosphere is very poor, which seriously limits the interest of its study. Please comment on it.
Lines 93 to 94. The authors report that “In Aug. 2021, 10 months after the last irrigation of GUMT319, soil samples (one kg 93 each) were randomly collected from locations in the grape orchard at a depth of 0–20 cm 94 near the rhizosphere of GUMT319-treated or untreated grape plants”. If the chemical and microbiological characterization of the soil was carried out 10 months after the last inoculation, it is very difficult to attribute any result to the activity of the bacteria. It must be remembered that the effective use of PGPRs implies repeated applications during each agricultural cycle. Thus, the reported differences in pH and microbiological diversity are hardly attributable to the activity of GUMT139.
Lines 104 to 106. “The organic matter 104 (C, N, and S) and elemental composition (wt %) in soil samples were determined following the method established by Hsieh (2007).” Hsieh (2007) reports a methodology to quantify C, N and S but not from soil samples, so the authors should include a detailed description of the methodology used. On the other hand, the methods to quantify hydrolysable alkaline nitrogen K, P, vitamin C, sugar/acid ratio, soil organic matter are not reported.
Lines 134 to 137. “After one year of constant treatment, the morphology of the GUMT319 inoculated grape plants was significantly different from the control plants. The ear length of the GUMT319-treated grape plants is longer than the controls, and the roots of GUMT319-treated plants had more branches and root hairs than the control plants.” Results not shown?
The methodology lacks a detailed description of the agronomic management of the vineyard. Crop fertilization/nutrition, irrigation, pest and disease control, grape variety, harvest, plot size, distribution of treatments, repetitions, etc.
Fig 5 B. Differences between treatments do not appear to be significant
Fig 6. KO Labels are diffused and need to be improved
Lines 219 to 227. The effect of pH is approached very simplistically since low soil pH reduces the supply of secondary macronutrients, whereas higher pH restricts soil micronutrient availability. Concluding that pH differences in the soil explain the greater growth of plants and fruits is very risky.
Lines 235 to 243. The authors try to attribute the increase in the biomass of the plants and fruits treated with GUMT139 to the fact that the microbial community has a greater capacity for Leucine production. This is a clear example of the lack of scientific rigor since all proteins contain leucine, not only those related to defense against pathogens. Furthermore, an increase in leucine production in the microbial community does not necessarily imply that this amino acid would be more available to the plant.
Conclusion: “In summary, this study demonstrated that the Bacillus velezensis strain GUMT319 is at 245 PGPR”. There are no results in this work that support this conclusion. On the other hand, it seems that this has already been previously demonstrated, as the authors comment (without including the reference): “B. velezensis strain GUMT319 is a PGPR rhizobacteria isolated from Guizhou province (Ding et al., 2021).”
Author Response
Response to reviewer 1
Thanks for reviewer’s comments.This manuscript indeed lacks the most direct experiment on Bacillus velezensis strain GUMT319 directly increased the yield of grape. The most important result of this manuscript is that we found and reported the yield increasing of grape when treated with Bacillus velezensis strain GUMT319, and speculate on possible associations between that Bacillus velezensis strain GUMT319 and the yield increasing of the grape, based on the high-throughput sequencing results and evaluation of the properties of the soil. Based on our experimental design, the main difference between the experimental group and the control group is whether Bacillus velezensis strain GUMT319 was used, so the possibility that the yield increase of grape is actually associated with Bacillus velezensis strain GUMT319 is extremely high.
Thanks for your reminding, the paper published by Ding will be added properly in the latest version of our manuscript. (Ding H , Mo W , Yu S , et al. Whole Genome Sequence of Bacillus velezensis Strain GUMT319: A Potential Biocontrol Agent Against Tobacco Black Shank Disease[J]. Frontiers in Microbiology, 2021, 12:658113.)
Thanks for your comment, you are right I miscalculated this, the real applied dose of GUMT319 we used here is
Thanks for your comment, the reason why we chose to investigate the soil microbiome and soil properties after 10 months of the last treatment is that we want to exclude the influence of the GUMT319 self on the result of soil microbiome, If we do not take this action, it is possible that the residual GUMT319 in the soil will affect the accuracy of our high throughput sequencing results of microbiome biodiversity. As there is only one major variable factor between samples, that is whether they were treated with GUMT319, we’d easily get the conclusion that the changes of grape is caused by GUMT319.
Thanks for your comment, as the evaluation of the physical and chemical properties is carried out by the Agricultural Resources and Environment Center of Guizhou Academy of Agricultural Sciences, we will check again with the testing institution and improve this part of the analysis method.
Thanks for your comment, the differences were shown in figure 1.
Thanks for your comment, the methodology of the agronomic management will be described in our latest version of manuscript.
Thanks for your comment, the differences is indeed are not so significant, but it will be more and more significant after constantly treatment of the GUMT319, actually we still carrying on this experiment, the three years and four years data will be analyzed and published in the future.
Thanks for your comment, the KO figure will be redrawn.
Thanks for your comment, the pH can alter the nutrient absorbance of the plant is a fact, we just give an explanation of the yield increase on GUMT319 treated grape based on our data and previous studies.
Thanks for your comment, “The alkaline hydrolyzable nitrogen and available potassium were beneficial to the synthesis of organic matter in the grape plants, which may have led to higher yields in the grape plants treated with GUMT319. The GUMT319 treatment apparently increased the nutrient absorption activity of grape plants, but the specific action involved requires further study. ” It is very logical that the nitrogen and potassium are important components in organic synthesis, there absorbance could affect the synthesis of organic matter in the plant is reasonable, and it is just a conclusion based on the result we got in figure 3, so we put it in the discussion part but not the result section.
Reviewer 2 Report
The paper can be accepted without any further changes.
Author Response
Thanks for your comment, we will improve the quality of the english writing of this manuscript according to your request.
Reviewer 3 Report
The article ¨ Bacillus velezensis strain GUMT319 reshapes microbiome biodiversity and increases grape yields¨ by Chen et al. is a good work that can be publishable in Biology journal where the authors propose the role of the biocontrol B. velezensis strain GUMT319 as a plant growth-stimulating bacterium. However, it requires some minor changes and be edited by a native English speaker or any professional editing services. The whole manuscript requires a lot of work in structure and flow.
Minor changes:
Line 14, correct rihizobacteria.
Line 34, The first paragraph o fthe Introduction section must be rewritten. Two time the wuthros star the phrase with: ¨Rhizosphere bacteria…¨ Therefore, the sentence needs reading fluency (This can be improved by any professional editing services)
Line 36, the reference by Kloepper et al., 1980 is okay, but too old. The same for other references from 1991!!! These works have more than 30 years old. Please update them.
Figures 2 and 3 need to be re-constructed again, too wide spaces between the bars. The letters are also too small and cannot be read (Well, maybe with a magnifying glass).
Figure 4, please correct the legends, particularly ¨Nosignifi¨. What is that abbreviation???
Figure 5, what analysis is Shannon idex? Or is it, Shannon index?
The discussion needs to be improved, it is too short with very few citations or works to compare with
References need to be corrected and double checked again. Some scientific names are incorrectly written.
Author Response
Thanks for your comment, the word rhizobacteria is corrected.
Thanks for your comment, the first paragraph will be rewritten as you suggested.
Thanks for your comment, the old references will be updated following your instructions.
Thanks for your comment, figure 2 and figure 3 will be reconstructed as you suggested.
Thanks for your comment, nosignifi here means the microbes which have no significant changes between the GUMT319 treated and untreated soil, the legend of figure 4 will be corrected follow your instructions.
Thanks for your comment, it should be index here in figure 5 and I misspelled it in the figure, this will be corrected in our latest version of manuscript.
Thanks for your comment, the discussion, references, and writing will be improved in the latest version of our manuscript following your advice.
Round 2
Reviewer 1 Report
Authors have not properly addressed the reviewers comments. As an example:
Response: Thanks for your comment, the reason why we chose to investigate the soil microbiome and soil properties after 10 months of the last treatment is that we want to exclude the influence of the GUMT319 self on the result of soil microbiome, If we do not take this action, it is possible that the residual GUMT319 in the soil will affect the accuracy of our high throughput sequencing results of microbiome biodiversity. As there is only one major variable factor between samples, that is whether they were treated with GUMT319, we’d easily get the conclusion that the changes of grape is caused by GUMT319.
The reviewer must insist. How the authors can attribute to GUMT319 a response that was evaluated 10 months after treatment. Many uncontrolled external factors could have affected the responses.
Example 2.
Response: Thanks for your comment, the methodology of the agronomic management will be described in our latest version of manuscript.
The agronomic management of the methodology was not included in new version
Author Response
All the answers to reviewer 1 were listed in the PDF file.

Round 3
Reviewer 1 Report
Authors have not properly addressed the main issues regarding the effect of GUMT 319 on grape quality and soil microbiome. In fact additional experiments are needed.
Author Response
Thank you very much for your valuable comments.